**Data Availability Statement:** We have an institutional data policy that restricts the use of data from third parts and its open access via internet,

# Individual and familial factors predict formation and improvement of adolescents' academic expectations: A longitudinal study in Sweden

**Melody Almroth**[1]*, **Krisztina D. László**[1], **Kyriaki Kosidou**[1,2], **Maria Rosaria Galanti**[1,2]

**1** Department of Global Public Health, Karolinska Institutet, Stockholm, Sweden, **2** Centre for Epidemiology and Community Medicine (CES), Stockholm Country Council, Stockholm, Sweden

* melody.almroth@ki.se

## Abstract

### Background

Adolescents' high academic expectations predict future health and successful societal integration. Yet, little is known about which factors may promote adolescents' expectations of their future education and academic achievement.

### Aims

To explore whether potentially modifiable factors such as parents' engagement and expectations regarding their child's education; or student individual factors such as school engagement, academic achievement, sense of identity, and positive mental health predict positive development of academic expectations in early adolescence.

### Methods

A longitudinal study of 3,203 adolescents and their parents was conducted with information collected between 7th grade (13 years of age) and 9th grade (16 years of age). Parental and adolescents' own academic expectations and engagement in school, academic achievement, identity synthesis, and mental health were self-reported in annual questionnaires. We used logistic regression to analyze the associations between the aforementioned factors and two binary outcomes related to changes in expectations from 7th to 9th grade: A. resolved uncertainty regarding own academic expectations; B. raised academic expectations.

### Results

Student engagement, and higher academic grades predicted both resolved uncertainty in expectations and raised academic expectations. Higher parental involvement in education was related to resolved uncertainty, while high parental expectations were related to raised student expectations. Identity synthesis and mental health did not appear to predict either outcome.

unless clearly stated in the protocol and explicitly allowed by the Ethical Committee, i.e. included in the consent that individuals release at the moment of entering the study. This was not the case in the KUPOL study; therefore, we cannot make a minimum dataset public. Data is available through application at kupolstudien.se, and the ethics committee can be contacted at registrator@etikprovning.se.

**Funding:** The KUPOL study is funded by a grant (nr 259-2012-48) containing funds from the Swedish Research Council Formas, The Swedish Research Council for Health, Working Life and Welfare, and The Swedish Research Council-Vetenskapsrådet awarded to MRG. The funders had no role in study design, data collection and analysis, decision to publish, or preparation of the manuscript.

**Competing interests:** The authors have declared that no competing interests exist.

## Conclusion

Our findings indicate that a supportive parental attitude concerning their child's education during adolescence, student engagement, and positive progressions in academic achievements may contribute to a positive development of academic expectations, thus to positive educational trajectories.

## Introduction

Studies have consistently linked adolescents' high academic expectations or aspirations with higher academic achievement [1–3], and more positive mental health [4–6]. A better understanding of the potentially modifiable factors that influence adolescents' academic expectations may be important for the development of potential educational and mental health trajectories.

Although research on the determinants of academic expectations is relatively scant, several previous studies have determined some important potentially modifiable antecedents of academic expectations or aspirations. Parental attitudes towards education may be of particular importance for adolescents' development of academic expectations. In fact, parental expectations for their children appear to be one of the most consistent predictors of adolescents' own academic expectations [7–9]. Parental involvement in their children's education, however, is less clearly understood. While this has been described as one of the most important predictors of academic expectations [10], others have reported null findings [11, 12].

According to the Expectancy-Value theory, academic motivation and achievement are predicted by the combination of ability self-concept and the subjective value put on certain tasks [13]. Previous academic achievement is likely to inform ability self-concept, and academic engagement may be an important aspect in the valuing of education and schoolwork. Previous studies have found intrinsic motivation towards schoolwork [7] and own academic ability or achievement to be important contributors to adolescent academic expectations [7, 8]. Some studies, however, cast doubt on the extent to which adolescents actually adjust their academic expectations according to their own ability or achievement [14, 15]. Mental health may also be an important aspect in relation to the development of academic expectations and goals and has previously been found to predict academic expectations [16].

Identity development is another factor which is closely linked to educational and vocational choices and future planning in general [17]. According to Erikson's psychological stage theory, adolescence is the developmental stage where young people primarily face conflicts between identity synthesis and confusion. It is important at this stage for them to establish a sense of who they are and what their goals are [18]. Though identity formation is a very individual process, at least one school-based intervention has succeeded in modifying identity development [19]. Few quantitative studies have investigated aspects of identity formation in relation to academic expectations.

During adolescence, academic expectations from external sources (typically parents and teachers) often become more demanding, while intrinsic motivation in school [20] as well as academic self-concept [21] tend to deteriorate. At the same time, specific goals and future planning are likely to become more definite, as decisions become closer and more central in the lives of adolescents [8]. Along with increasing rational thinking, some adolescents may adjust their expectations to more realistic or achievable goals over time. Previous studies have come to conflicting conclusions about whether own expectations on average tend to increase [22], decrease [23] or stay the same [10, 14] during early adolescence.

Our understanding of potentially malleable factors that influence the development of adolescent academic expectations, however, remains poor. For instance, few studies have addressed the question of which factors explain positive *changes* in academic expectations during the critical time of adolescence. One particularly interesting change at this age is the transition between uncertain expectations to more definite visions about own future educational development, a topic often neglected in this type of study. On one hand, uncertainty in early adolescence may be a natural response related to healthy identity formation and adaptiveness [24]. On the other hand, prolonged uncertainty may lead to delayed goal setting and subsequent undesirable outcomes such as poor academic performance and attainments later on [7]. A better understanding of which factors influence an improvement in academic expectations before the transition to high school, including resolving uncertainty would be important in order to design consistent strategies to promote positive adolescent educational goals.

In Sweden, 9th grade (age 16) marks the end of compulsory school. At this point the students choose whether or not they want to attend upper-secondary education and if so, whether they would like to follow a vocational track, or a theoretical track, this latter often in preparation for university studies [25]. The time between 7th and 9th grade is therefore particularly critical for the development of educational plans in the Swedish context.

This study aims to explore potentially modifiable factors in the family environment and at the individual adolescent's level that may promote positive change in academic expectations during the final block of the Swedish compulsory school, i.e. between 7th and 9th grade. Specifically, we explored whether parental involvement with school, parental expectations, student engagement, academic grades, sense of identity, and mental health are associated with an improvement in academic expectations or with a resolution of uncertainty in these expectations.

## Methods

### Study population and design

This study is part of the KUPOL cohort study (a Swedish acronym for Knowledge about Young People's Mental Health and Learning). Details of the study design and the data collection methods have previously been published [26] and will be summarized here. One hundred and one schools (19% of those eligible) were recruited from southern and central Sweden. Within these schools, the parents of 3,959 students (32% of the eligible) gave informed consent for their children's participation in the study during the 2013–2014 or 2014–2015 academic years. Of the students allowed to participate, 3,671 answered the baseline questionnaire in the 7th grade (age 13). Yearly follow-up surveys were conducted in the 8th and 9th grade using similar questionnaires. There were 3,203 adolescents who participated both in the baseline and in the second follow up survey. Parents of the index adolescents also answered a similar questionnaire each year with 3,645 answering the baseline questionnaire and 2,977 answering at both the baseline and 2nd follow-up.

The KUPOL study was approved by the Stockholm Ethics Review Board (reference numbers: 2012/1904-31/1 and 2016/1280-32).

### Measures

**Outcome.** We measured academic expectations by asking adolescents both in 7th grade and in 9th grade how far they thought they would go in their education with the possible responses; "I don't know", "high school-vocational track", "high school-theoretical track", and "university". From these answers we derived two outcomes representing positive changes in academic expectations in two sub-populations. We first considered those with uncertain

expectations in the 7[th] grade (N = 1,429) and compared those who remained uncertain with those who endorsed a specific expectation in the 9[th] grade. Next, we considered those who reported some definite expectation in the 7[th] grade (N = 595) and compared those who raised their expectations between the 7[th] and the 9[th] grade with those who did not. We excluded those participants who reported university expectations at baseline (N = 1,017) from the main analysis, as it was not possible for them to experience any upward development in expectations. However, in secondary analyses we compared those who lowered their expectations between 7[th] and 9[th] grade with those who sustained or raised their expectations (N = 1,577).

**Predictors.** Parental involvement in their child's education was measured using the child-reported Family Support for Learning (FSL) subscale of the validated Student Engagement Instrument [27]. This subscale measures how engaged and supportive children feel that their families are concerning school life using four items with five response alternatives ranging from "strongly agree" to "strongly disagree". The total score is calculated as a mean of the four items [28] where a higher score indicates more family engagement in school. The Cronbach's alpha was .79 in this sample.

Parental academic expectations for their children were measured by asking how far parents expected their children to go in school. We categorized this variable as "university expectations" vs. "less than university expectations".

Adolescent engagement with school was measured using the Future aspirations and Goals subscale of the Student Engagement Instrument. This subscale includes five items related to future academic plans, general future prospects, and the importance put on school. A sample item is "School is important for achieving my future goals". Items are answered on a five-point scale ranging from "strongly disagree" to "strongly agree". The scale score is an average of the five items with a higher score indicating higher engagement. The Cronbach's alpha was .79 in this population.

Academic grades obtained during the 7[th] grade were retrospectively reported by the adolescent in either 8[th] or 9[th] grade for the subjects Swedish, English, and mathematics. Each grade was represented by a numeric value that was then added across subjects to derive a total score (a higher score indicating higher academic achievement).

Identity synthesis was assessed using five items from the Identity subscale of the Erikson Psychological Stage Inventory. This subscale indicates how secure the child feels with his or her own identity and has previously been validated [29]. The score was calculated as a mean of the five items, with a higher score indicating stronger identity synthesis. The Cronbach's alpha for this scale was .71.

Mental health was self-reported by the adolescent using two measures. The well validated Center for Epidemiological Studies Depression scale for Children (CES-DC) [30] was used to measure depressive symptoms. This scale assesses how often depressive symptoms have occurred during the last week. We used the cutoff point of 30 recommended for Swedish adolescents to separate normal and high scores [31]. The Strengths and Difficulties Questionnaire was used to measure multiple dimensions of mental health [32]. This scale assesses symptoms related to emotional problems, peer problems, conduct problems, and hyperactivity during the last six months. The recommended cutoff point of 18 was used in order to separate normal and high scores [33].

All predictors were measured at baseline when adolescents were in the 7[th] grade (age 13) except for academic grades which were reported retrospectively.

**Covariates.** Some additional variables were considered as potential confounders because they could theoretically determine both academic expectations and their predictors. These were: adolescents' gender; family living arrangement (living with two parents or one or neither of them), derived from the child's questionnaire; parents' education and country of birth,

obtained from the parents' questionnaire and categorized as "at least one parent with university education vs. neither parent with university education" and "both parents born in Sweden vs. one parent born outside of Sweden" respectively. Being female, living in a two-parent family, having a higher SES, and identifying as an ethnicity other than white have all previously been found to be related to having higher academic expectations [34]. These factors are also likely to be related to parental and own engagement in school, parental expectations, academic achievement, sense of identity and mental health.

## Statistical analysis

We analyzed baseline characteristics of the study population separately according to the two main outcome variables (resolved uncertainty and raised expectations) using chi-square tests for categorical variables and Kruskall-Wallis tests for continuous variables.

We used logistic regression models to estimate: A. the likelihood of endorsing specific expectations in 9th grade rather than remaining undecided among those reporting undecided expectations at baseline. B. the likelihood of raising expectations between 7th and 9th among those who reported a definite expectation in 7th grade.

Models were adjusted for sex, living arrangement, parents' education, and parents' country of birth.

As a secondary analysis, we explored the additional outcome of lowering expectations from 7th to 9th grade compared to those who maintained or raised their expectations.

We used multi-level models to accommodate for clustering at the school level. Because estimates and model test statistics (-2 log likelihood) were nearly identical in the two models, results are presented ignoring the school-level cluster.

All analyses were conducted using SAS enterprise guide 7.1.

## Results

About two thirds of adolescents who did not endorse a definite academic expectation at baseline resolved their uncertainty by the 9th grade. Among those who reported definite academic expectations at baseline, 47% raised their expectations. Table 1 shows the distribution of 9th grade expectations according to 7th grade expectations.

Adolescents living with both parents were more likely to resolve their uncertainty. Median student engagement, parental support and grades were higher among those who resolved their uncertainty between 7th and 9th grade. Girls, adolescents with parents with university education, those whose parents expected them to go to university, and those with at least one parent born outside of Sweden were more likely to raise their expectations between 7th and 9th grade. Median grades in 7th grade and student engagement were higher among those who raised their expectations between 7th and 9th grade. (Table 2).

Higher parental support was associated with a greater odds of transition to defined expectations (adjusted OR 1.37 95% CI 1.12–1.67), while high parental expectations were not. Student engagement in school was also associated with a significant increase in the odds of resolving uncertainty in expectations (OR 1.29 95% CI 1.09–1.54). Higher academic grades were associated with greater odds of resolving uncertainty (OR 1.03 95% CI 1.01–1.04). Identity synthesis and the absence of problematic depressive or mental health symptoms did not appear to be significant predictors of resolving uncertainty (Table 3).

Parental university expectations were associated with greater odds of their children raising their own academic expectations (OR 3.14 95% CI 2.06–4.79), as was adolescents engagement in school (OR 1.53 95% CI 1.16–2.04) and having higher academic grades (OR 1.05 95% CI 1.03–1.07). Parental involvement in their child's education, identity synthesis, and absence of

**Table 1. Expectations reported in the 9th grade according to 7th grade expectations.**

| Expectations 7th grade | Expectations 9th grade | | | | |
|---|---|---|---|---|---|
| | I don't know | High school vocational | High school theoretical | University | Total |
| | N (%) | N (%) | N (%) | N (%) | |
| I don't know | 472 (33) | 229 (16) | 248 (17) | 480 (34) | 1429 (100) |
| Highschool vocational | 52 (20) | 82 (31) | 72 (28) | 55 (21) | 261 (100) |
| High school theoretical | 51 (15) | 49 (15) | 84 (25) | 150 (45) | 334 (100) |
| University | 91 (9) | 35 (4) | 71 (7) | 785 (8+) | 982 (100) |
| Total | 666 | 395 | 475 | 1470 | 3006 |

depressive or emotional and behavioral symptoms did not appear to be important predictors of raising academic expectations (Table 4).

Secondary analyses of lowered expectations revealed a similar pattern, with higher parental expectations, student engagement, and academic grades all being significant predictors of a lower odds of adolescents lowering their expectations. Parental engagement, identity synthesis, and a normal score on the Strengths and Difficulties Questionnaire did not significantly predict lowered expectations. However, having a normal score on the CES-DC depression scale was associated with a decreased odds of lowering expectations (S1 Table).

**Table 2. Baseline characteristics and change in aspirations between 7th and 9th grade among those with undecided expectations and those with decided expectations below university in 7th grade.**

| | | Uncertain expectations at baseline | | | Decided expectations at baseline | | |
|---|---|---|---|---|---|---|---|
| | | Sustained uncertainty | Uncertainty resolved | | No raised expectations | Raised expectations | |
| | | N = 472 | N = 957 | | N = 318 | N = 277 | |
| | | N (%) | N (%) | P-value[a] | N (%) | N (%) | P-value[a] |
| Sex of child | Girls | 235 (32) | 506 (68) | | 144 (49) | 152 (51) | |
| | Boys | 237 (34) | 451 (66) | 0.2723 | 174 (58) | 125 (42) | 0.0196 |
| Living arrangement | Both parents | 434 (32) | 907 (68) | | 291 (53) | 259 (47) | |
| | Not with both parents | 37 (43) | 49 (57) | 0.0416 | 27 (60) | 18 (40) | 0.3592 |
| Parental education | University education | 292 (31) | 643 (69) | | 193 (49) | 202 (51) | |
| | No university education | 169 (36) | 303 (64) | 0.0843 | 125 (63) | 73 (37) | 0.0010 |
| Parents' birth country | Both Sweden | 357 (33) | 733 (67) | | 253 (55) | 209 (45) | |
| | At least one outside of Sweden | 85 (35) | 158 (65) | 0.5049 | 39 (43) | 52 (57) | 0.0376 |
| Parents expectations | University | 241 (31) | 547 (69) | | 136 (42) | 188 (58) | |
| | Less than university | 182 (35) | 333 (65) | 0.0730 | 156 (72) | 61 (28) | < .0001 |
| CES-DC | Non-problematic score | 424 (33) | 854 (67) | | 268 (53) | 242 (47) | |
| | High score | 42 (33) | 87 (67) | 0.8868 | 46 (57) | 35 (43) | 0.4774 |
| SDQ total | Non-problematic score | 422 (33) | 865 (67) | | 277 (53) | 247 (47) | |
| | High score | 43 (34) | 82 (66) | 0.7145 | 38 (58) | 28 (42) | 0.4695 |
| | | Median | Median | | Median | Median | P-value[a] |
| Future aspirations and goals | | 4.20 | 4.40 | 0.0013 | 4.40 | 4.60 | 0.0002 |
| Family support for learning | | 4.75 | 5.00 | 0.0040 | 4.75 | 4.75 | 0.2876 |
| Grades | | 40.00 | 42.50 | < .0001 | 40.00 | 45.00 | < .0001 |
| Identity synthesis | | 3.83 | 3.83 | 0.9899 | 3.83 | 3.67 | 0.3499 |

CES-DC = Center for Epidemiological Studies Depression scale for Children, SDQ = Strengths and Difficulties Questionnaire.

[a] P-values correspond to chi-square tests for categorical variables and Kruskal-Wallis tests for continuous variables.

**Table 3. Odds ratios and 95% confidence intervals for transition to definite expectations between 7th and 9th grade among adolescents who reported no definite expectations at baseline, according to family and individual predictors (N = 1,429).**

| Variable | Model 1[a] | Model 2[b] |
|---|---|---|
| | OR (95% CI) | OR (95% CI) |
| Family support for learning (continuous) | 1.31 (1.07–1.58) | 1.37 (1.12–1.67) |
| Parental expectations university vs. other | 1.24 (0.98–1.57) | 1.17 (0.90–1.52) |
| Future aspirations and goals (continuous) | 1.27 (1.08–1.51) | 1.29 (1.09–1.54) |
| Grades (continuous) | 1.03 (1.02–1.04) | 1.03 (1.01–1.04) |
| Identity synthesis (continuous) | 1.00 (0.87–1.14) | 1.06 (0.91–1.23) |
| CES-DC (ref = high score for depressive symptoms) | 0.97 (0.66–1.43) | 1.08 (0.72–1.64) |
| SDQ total (ref = high score for total difficulties) | 1.08 (0.73–1.58) | 1.14 (0.76–1.70) |

OR = odds ratio, CI = confidence interval, CES-DC = Center for Epidemiological Studies Depression scale for Children, SDQ = Strengths and Difficulties Questionnaire.

[a] Model 1 is unadjusted.

[b] Model 2 is adjusted for child's gender, living arrangement, parental education, and parents' country of birth.

## Discussion

In this large longitudinal study, we found that both parental attitudes and adolescents' engagement and achievements were associated with an improvement in the future expectations about own education. Specifically, higher parental involvement in education, higher student engagement, and academic grades predicted whether students who in the 7th grade could not endorse any definite expectations would resolve this uncertainty two years later. Likewise, higher parental academic expectations, higher student engagement and higher academic grades predicted raised academic expectations between 7th and 9th grade. However, psychological factors at a typically individual level such as identity synthesis and positive mental health were not longitudinally associated with positive change in academic expectations in this cohort.

Previous studies have consistently found parental expectations to predict adolescents' own expectations [7–9]. However, studies investigating the association between parental involvement in or support for learning and their children's academic expectations tend to come to

**Table 4. Odds ratios and 95% confidence intervals of raised own academic expectations among adolescents reporting a definite academic expectation at baseline according to family and individual predictors (N = 595).**

| Variable | Model 1[a] | Model 2[b] |
|---|---|---|
| | OR (95% CI) | OR (95% CI) |
| Family support for learning (continuous) | 1.03 (0.77–1.36) | 1.09 (0.81–1.46) |
| Parental expectations university vs. other | 3.54 (2.44–5.11) | 3.14 (2.06–4.79) |
| Future aspirations and goals (continuous) | 1.67 (1.27–2.18) | 1.53 (1.16–2.04) |
| Grades (continuous) | 1.06 (1.04–1.08) | 1.05 (1.03–1.07) |
| Identity synthesis (continuous) | 0.93 (0.76–1.13) | 1.02 (0.81–1.29) |
| CES-DC (ref = high score for depressive symptoms) | 1.19 (0.74–1.90) | 1.34 (0.80–2.24) |
| SDQ total (ref = high score for total difficulties) | 1.21 (0.72–2.03) | 1.34 (0.77–2.34) |

OR = odds ratio, CI = confidence interval, CES-DC = Center for Epidemiological Studies Depression scale for Children, SDQ = Strengths and Difficulties Questionnaire.

[a] Model 1 is unadjusted

[b] Model 2 is adjusted for child's gender, living arrangement, parental education, and parents' country of birth.

different conclusions. Several found parental involvement in education to be an important predictor of adolescent academic expectations [9, 10, 35] while others reported null associations [11, 12]. However, measures of parental engagement vary between studies. For instance, some of these studies measured parental involvement in terms of how often parents met with teachers or attended meetings. This behavior may not be an uncontroversial measure of engagement, since it is likely to also reflect children's difficulties in school, thus hampering a direct interpretation of this association. We measured parental involvement through the children's perception of their parents' engagement and communication related to schoolwork. Interestingly, in this study parental involvement in the child's education appeared especially important for resolving uncertainty in expectations, while parental academic expectations appeared especially important for predicting a raise in adolescents' academic expectations. It is possible that parental non-directive engagement in the adolescents' education promotes the process of autonomous decision making and goal setting rather than the absolute level of these goals [36]. On the other hand, parents communicating high academic expectations may prompt their children to strive for higher specific goals.

Higher *own* engagement in school has been found to be related to academic expectations in qualitative [37] and quantitative studies [38], to the point that expectations and engagement are sometimes measured as a single construct of "engagement with educational goals" [39]. Our findings support that engagement in school predicts a positive shift in academic expectations during this phase of adolescents.

A bi-directional relationship between academic achievement and academic expectations has previously been hypothesized. Some studies have found that adolescents adjust their expectations based on their own achievement. For instance, one study found that adolescents' high achievement predicted higher expectations [8], but other studies did not find such a relationship. In one study, only very large changes in average grades predicted a change in expectations [14]. In another study the majority of adolescents did not adjust their expectations to reflect their current achievements [15]. Our findings present some support for the hypothesis that own achievement is important in inspiring higher expectations. This is in line with Expectancy-Value theory, where achievement and motivation are predicted by ability self-concept (previous achievement) and subjective task value (engagement) [13]. It is likely that academic achievement and expectations reinforce each other over time indicating that adolescents adjust their expectations according to their own achievements, and that improving expectations is likely to also improve future achievements.

Sense of identity has sometimes been described as closely related to the formation of academic and career goals [40], but a long period of exploration takes place before identity commitments are made [41]. In our study, a greater sense of identity synthesis was not related to resolving uncertainty of expectations or to raised expectations. It may be that both identity and academic expectations develop simultaneously rather than early adolescent identity development predicting later changes in expectations. Though identity formation is supposed to take place during adolescence, it is unclear at exactly what age identity synthesis should be achieved [42]. The adolescents in our sample may have been too young at baseline to convey useful information about their identity development. It is also possible that identity formation and academic expectations actually represent two different and independent processes. Adolescents may be able to separate their specific academic goals from who they feel they are as a person overall.

Previous studies have found that measures of mental health or well-being tend to be associated with higher educational expectations or aspirations [6, 12, 16, 43], though some of these studies were cross-sectional and offered different explanations of the causal direction of these associations [6, 43]. In our study, mental health measures did not predict a positive change in

academic expectations. It may be that the relationship between mental well-being and positive expectations is already established at an early age.

Inequalities in Swedish education are growing in terms of an increasing gap between the highest and lowest achievers in general, as well as a growing achievement gap between those with high and low socioeconomic status [44]. Those who are thriving in the current academic system may be particularly adaptive individuals with supportive families and more optimistic perceptions of their potential opportunities for the future.

Several limitations of our study need to be considered. The original cohort had a low response rate at the individual level (ca 30%) and was rather selected with an over-representation of highly educated parents and an under-representations of parents born outside of Sweden [26]. For this reason, the generalizability of these results may be limited. Additionally, we lacked finer insights in the question measuring academic expectations, as university was the highest level of education possible for the student to report. We also only had a select number of items from the Erikson Psychological Stage Inventory, and thus may not have captured all aspects of identity development. Relying on retrospectively reported grades also led to poorer quality data in this variable. Finally, unmeasured confounding cannot be ruled out, for instance parents' mental health, the child's cognitive ability, or personality traits such as optimism.

## Conclusion

This study found that potentially modifiable factors at the family and at the individual level such as parental engagement in education, parental academic expectations, and adolescent engagement and achievement in school are predictors of a positive development in academic expectations during the final years of Swedish compulsory school. These findings point toward the high value of school-family communication, which is likely to impact on parental engagement in children's education and on how parental expectations are formed and communicated. In turn these processes would facilitate adolescents' identification of their academic goals and ultimately their social, psychological, and educational trajectories.

## Supporting information

**S1 Table. Odds ratios and 95% confidence intervals for lowering own academic expectations among adolescents reporting a definite academic expectation at baseline according to family and individual predictors (N = 1,577).**
(DOCX)

## Acknowledgments

We would like to thank the participating adolescents, families, and schools and the executive team of the KUPOL study who make our research possible.

## Author Contributions

**Conceptualization:** Melody Almroth, Krisztina D. László, Kyriaki Kosidou, Maria Rosaria Galanti.

**Data curation:** Maria Rosaria Galanti.

**Formal analysis:** Melody Almroth.

**Funding acquisition:** Maria Rosaria Galanti.

**Investigation:** Maria Rosaria Galanti.

**Methodology:** Melody Almroth, Krisztina D. László, Kyriaki Kosidou, Maria Rosaria Galanti.

**Project administration:** Maria Rosaria Galanti.

**Resources:** Maria Rosaria Galanti.

**Software:** Melody Almroth.

**Supervision:** Krisztina D. László, Kyriaki Kosidou, Maria Rosaria Galanti.

**Writing – original draft:** Melody Almroth.

**Writing – review & editing:** Krisztina D. László, Kyriaki Kosidou, Maria Rosaria Galanti.

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
