## [Decision Letter · Decision Letter 0]

22 Nov 2019

PONE-D-19-23059

Individual and familial factors predict formation and improvement of adolescents’ academic expectations: a longitudinal study in Sweden

PLOS ONE

Dear Ms. Almroth,

Thank you for submitting your manuscript to PLOS ONE. After careful consideration, we feel that it has merit but does not fully meet PLOS ONE’s publication criteria as it currently stands. Therefore, we invite you to submit a revised version of the manuscript that addresses the points raised during the review process.

Please find the reviewers report below this email and address their comments which would be helpful in preparing the revised manuscript. They all agree that is an important and well written study but they have some queries regarding data and analysis. 

We would appreciate receiving your revised manuscript by Jan 06 2020 11:59PM. To enhance the reproducibility of your results, we recommend that if applicable you deposit your laboratory protocols in protocols.io, where a protocol can be assigned its own identifier (DOI) such that it can be cited independently in the future. For instructions see: http://journals.plos.org/plosone/s/submission-guidelines#loc-laboratory-protocols

We look forward to receiving your revised manuscript.

Kind regards,

Muhammad A. Z. Mughal

Academic Editor

PLOS ONE

Journal Requirements:

*In your revised cover letter, please address the following prompts:

Reviewers' comments:

Reviewer's Responses to Questions

**Comments to the Author**

1. Is the manuscript technically sound, and do the data support the conclusions?

Reviewer #1: Yes

Reviewer #2: Partly

Reviewer #3: Partly

2. Has the statistical analysis been performed appropriately and rigorously? 

Reviewer #1: Yes

Reviewer #2: I Don't Know

Reviewer #3: Yes

3. Have the authors made all data underlying the findings in their manuscript fully available?

Reviewer #1: No

Reviewer #2: Yes

Reviewer #3: Yes

4. Is the manuscript presented in an intelligible fashion and written in standard English?

Reviewer #1: Yes

Reviewer #2: Yes

Reviewer #3: Yes

5. Review Comments to the Author

Reviewer #1: A fair study with rich data. Here I have a few comments to, hopefully, improve the manuscript.

1) Please make a table / figure to indicate the statistics on academic expectation improvement, no change, and decrease.

2) Related to (1)—this manuscript only considered positive factors. I suggest investigating the factors associated with decrease of academic expectation, too.

3) It seems factors in Table 2 were put into the model separately. How about putting them all into the same model to find which remain significant?

4) The language is fine in the main text; in comparison, the Abstract seemed coarse. For example, the second sentence “This is unfortunate xxx”, which is out of line for a research paper; in the third sentence the first prep. “in” should be removed. Please refine the Abstract.

Reviewer #2: This submission reports on a project that examined change in young adolescent expectations regarding future education. Two kinds of variables were assessed: family variables and individual student variables. The family variables included parental engagement with education and their educational expectations for their children. The individual variables included student engagement with education, their educational expectations, the extent of their identity work, academic achievement and mental health status. One of the nice things about this submission is the longitudinal design. Data were analysed using logistic regression

And the reported results were straightforward and unsurprising, and supported certain other findings in the literature. For example, what matters for resolving uncertainty in expectations and rising educational expectations? Student engagement and good grades. Also, higher parental involvement in education and parental expectations was related to rising expectations. Certain variables did not work out, e.g., mental health and identity synthesis did not predict educational expectations.

Although the results seem sensible some methodological decisions were made that give me pause. First, the variable Adolescent Engagement with School was measured with two items from the "future aspirations" and "goals" subscale(s) of the Student Engagement Inventory. A reader will want to know if there were items other than these two to be found on the SEI subscales and, if so, why weren't they included?

Second, it is a puzzle why the Likert-type scaling of this measure was transformed into a dichotomy. Insofar as the original scaling was on a continuum, what would be the advantage of ignoring the continuum to create the dichotomous variable---surely not to accommodate logistic regression. I should think that much information, and variability is lost ignoring the continuum.

Third, are we confident that young adolescents can retrospectively report their grades? This is not a big deal, but I wonder if more could be learned by NOT averaging the Swedish, English and mathematics grades into one academic achievement variable. I suspect that students' expectations of their future might hinge on how well they do in mathematics (for example) more than, say, Swedish, and, again, a potential clarifying finding is left unexplored.

Finally, I do worry about the proliferation of covariates. I am not as convinced as the authors' that they are included out of theoretical considerations (for no theory is offered).

Reviewer #3: The study explores whether potentially modifiable family factors, such as parental engagement and expectations regarding the education of their children, as well as individual factors such as school engagement, academic performance, sense of identity and mental health, predict the positive development of academic expectations in early adolescence. To do this, the authors use a longitudinal study with 3,204 adolescents and their parents. The data is collected through a questionnaire applied annually. For the analysis of the associations between the aforementioned factors, logistic regression is used, establishing: A. the student resolves the uncertainty regarding the academic expectations themselves; B. Increase in academic expectations between 7th and 9th grades. Concerning the results, school commitment and higher academic grades predicted the resolution of uncertainty in expectations, as well as an increase in academic expectations. Also, the participation of parents in the education of their children was related to the uncertainty resolved, while the expectations of the parents were related to the increase in expectations between 7th and 9th grade. Regarding identity and mental health, these variables failed to predict any of the results.

The strengths of the study lie fundamentally in the breadth of the sample and in the relevance of the established associations. Regarding the weaknesses of this study, although I have marked "yes" in question 2, I doubt the statistical analysis models used. In this sense, I wonder if it would not have been better to use a model of structural equations, specifying the use of developed indicators to capture the characteristics of categorical variables. The use of SEM would require a larger number of items of the Adolescent engagement with school scale (only 2 items are used).

Details:

Details:

Q. 7. The Alpha value is not provided in the Adolescent engagement with school Scale.

P. 12. The notes in Table 1 refer to the standard deviation (SD), but I cannot find that data in the table (nor in the other tables).

6. PLOS authors have the option to publish the peer review history of their article (what does this mean?). If published, this will include your full peer review and any attached files.

Reviewer #1: No

Reviewer #2: Yes: Daniel Lapsley

Reviewer #3: No

---

## [Author Response · Author response to Decision Letter 0]

20 Jan 2020

Reviewer 1

Comment 1: A fair study with rich data. Here I have a few comments to, hopefully, improve the manuscript. Please make a table / figure to indicate the statistics on academic expectation improvement, no change, and decrease.

Response: Thank you for the generous appraisal of our study and the comments. We have now included a table showing adolescents’ 9th grade expectations according to their expectations at baseline. It is therefore possible to see the academic expectations’ trajectory of all individuals in the study (page 10). 

Comment 2: Related to (1)—this manuscript only considered positive factors. I suggest investigating the factors associated with decrease of academic expectation, too.

Response: We have now included a supplementary analysis, where the outcome is conceptualized as lowered expectations versus sustained or raised expectations among students who from the start endorsed definite expectations. (page 7, 9, 14, and 24).

Comment 3: It seems factors in Table 2 were put into the model separately. How about putting them all into the same model to find which remain significant?

Response: Our goal was to look at the direct effects of potential predictors of positive changes in academic expectations while adjusting for potential confounding factors. These were chosen a priori based of their potential to be (causally) related to both the exposure and outcome but not laying in the causal pathway between the two. Combining all potential predictors into one model could result in a distorted interpretation of the effects of each variable due to over-adjustment on factors which do not have a clear potential to be confounding factors, or work as mediators. This adjustment could lead to spurious associations or to an underestimation of the total causal effect. 

Comment 4: The language is fine in the main text; in comparison, the Abstract seemed coarse. For example, the second sentence “This is unfortunate xxx”, which is out of line for a research paper; in the third sentence the first prep. “in” should be removed. Please refine the Abstract.

Response: We have now revised the abstract to improve language and clarity (page 2).

Reviewer 2

Comment 1: This submission reports on a project that examined change in young adolescent expectations regarding future education. Two kinds of variables were assessed: family variables and individual student variables. The family variables included parental engagement with education and their educational expectations for their children. The individual variables included student engagement with education, their educational expectations, the extent of their identity work, academic achievement and mental health status. One of the nice things about this submission is the longitudinal design. Data were analyzed using logistic regression

And the reported results were straightforward and unsurprising, and supported certain other findings in the literature. For example, what matters for resolving uncertainty in expectations and rising educational expectations? Student engagement and good grades. Also, higher parental involvement in education and parental expectations was related to rising expectations. Certain variables did not work out, e.g., mental health and identity synthesis did not predict educational expectations.

Response: Thank you for your accurate summary of our study and for taking the time to review it. 

Comment 2: Although the results seem sensible some methodological decisions were made that give me pause. First, the variable Adolescent Engagement with School was measured with two items from the "future aspirations" and "goals" subscale(s) of the Student Engagement Inventory. A reader will want to know if there were items other than these two to be found on the SEI subscales and, if so, why weren't they included?

Response: We originally omitted the items in the SEI that related directly to future academic plans because they did not seem to convey new information (e.g., that agreeing with the statement “I want to continue my education after high school” would almost certainly predict university expectations). Therefore, we originally chose to analyze only the two items that would best represent academic engagement. However, supplementary analyses revealed that the items related directly to future academic plans were not more correlated with the outcomes than the items on academic engagement. In the revised version, we used the entire scale (page 7).

Comment 3: Second, it is a puzzle why the Likert-type scaling of this measure was transformed into a dichotomy. Insofar as the original scaling was on a continuum, what would be the advantage of ignoring the continuum to create the dichotomous variable---surely not to accommodate logistic regression. I should think that much information, and variability is lost ignoring the continuum.

Response: This choice was originally made because the distribution of the answers was centrally skewed, i.e. there were very few who did not agree or strongly agree with either of the statements (around 5-7%). In addition, Likert scale response alternatives are ordinal/categorical, not continuous. Using the total scale score in the revised version, however, automatically solves this problem. 

Comment 4: Third, are we confident that young adolescents can retrospectively report their grades? This is not a big deal, but I wonder if more could be learned by NOT averaging the Swedish, English and mathematics grades into one academic achievement variable. I suspect that students' expectations of their future might hinge on how well they do in mathematics (for example) more than, say, Swedish, and, again, a potential clarifying finding is left unexplored.

Response: We agree that complete reliability cannot be expected when adolescents retrospectively reported their grades, a limitation that we now address in the corresponding section (Page 18). Indeed, among those who reported their 7th grade grades in both 8th and 9th grade, the correlation between these two time points was 0.82 for Swedish, 0.85 for English, and 0.86 for math indicating that there is not complete matching between these self-reports. Upon your request, we also looked at the effects of grades in each of the three subjects. We found the estimates for English to be slightly lower and math to be slightly higher for both outcomes, but confidence intervals were overlapping. We chose not to report these results in the revised manuscript.

Odds ratios and 95% confidence intervals for resolved uncertainty according to subject specific grades

 Unadjusted Adjusted

Swedish 1.07 (1.04-1.10) 1.06 (1.03-1.10)

English 1.04 (1.02-1.08) 1.05 (1.02-1.08)

Math 1.07 (1.04-1.10) 1.07 (1.04-1.10)

adjusted for child’s gender, living arrangement, parental education, and parents’ country of birth.

Odds ratios and 95% confidence intervals for raised expectations according to subject specific grades

 Unadjusted Adjusted

Swedish 1.14 (1.08-1.21) 1.11 (1.05-1.18)

English 1.10 (1.05-1.15) 1.08 (1.03-1.13)

Math 1.14 (1.09-1.20) 1.12 (1.07-1.17)

adjusted for child’s gender, living arrangement, parental education, and parents’ country of birth.

Comment 5: Finally, I do worry about the proliferation of covariates. I am not as convinced as the authors' that they are included out of theoretical considerations (for no theory is offered). 

Response: We have now added in the introduction theoretical perspectives in our choices of potential predictors (page 3) and further explanation of our choice of confounders (page 8-9). 

Reviewer 3

Comment 1: The study explores whether potentially modifiable family factors, such as parental engagement and expectations regarding the education of their children, as well as individual factors such as school engagement, academic performance, sense of identity and mental health, predict the positive development of academic expectations in early adolescence. To do this, the authors use a longitudinal study with 3,204 adolescents and their parents. The data is collected through a questionnaire applied annually. For the analysis of the associations between the aforementioned factors, logistic regression is used, establishing: A. the student resolves the uncertainty regarding the academic expectations themselves; B. Increase in academic expectations between 7th and 9th grades. Concerning the results, school commitment and higher academic grades predicted the resolution of uncertainty in expectations, as well as an increase in academic expectations. Also, the participation of parents in the education of their children was related to the uncertainty resolved, while the expectations of the parents were related to the increase in expectations between 7th and 9th grade. Regarding identity and mental health, these variables failed to predict any of the results.

Response: Thank you for taking the time to give feedback on our manuscript. 

Comment 2: The strengths of the study lie fundamentally in the breadth of the sample and in the relevance of the established associations. Regarding the weaknesses of this study, although I have marked "yes" in question 2, I doubt the statistical analysis models used. In this sense, I wonder if it would not have been better to use a model of structural equations, specifying the use of developed indicators to capture the characteristics of categorical variables. The use of SEM would require a larger number of items of the Adolescent engagement with school scale (only 2 items are used).

Response: Our interest was in the direct associations of the various predictors with positive changes in academic expectations. In this regard, we doubt that using SEM would convey a perceivable advantage compared to logistic regression.

Comment 3: Q. 7. The Alpha value is not provided in the Adolescent engagement with school Scale.

Response: We report the Cronbach’s alpha for this scale in the revised version (page 7). 

Comment 4: P. 12. The notes in Table 1 refer to the standard deviation (SD), but I cannot find that data in the table (nor in the other tables).

Response: Thank you for pointing out this mistake, which is now corrected.

---

## [Decision Letter · Decision Letter 1]

10 Feb 2020

Individual and familial factors predict formation and improvement of adolescents’ academic expectations: a longitudinal study in Sweden

PONE-D-19-23059R1

Dear Dr. Almroth,

We are pleased to inform you that your manuscript has been judged scientifically suitable for publication and will be formally accepted for publication once it complies with all outstanding technical requirements.

With kind regards,

Muhammad A. Z. Mughal

Academic Editor

PLOS ONE

Additional Editor Comments (optional):

Reviewers' comments:

Reviewer's Responses to Questions

**Comments to the Author**

1. If the authors have adequately addressed your comments raised in a previous round of review and you feel that this manuscript is now acceptable for publication, you may indicate that here to bypass the “Comments to the Author” section, enter your conflict of interest statement in the “Confidential to Editor” section, and submit your "Accept" recommendation.

Reviewer #1: All comments have been addressed

Reviewer #2: All comments have been addressed

2. Is the manuscript technically sound, and do the data support the conclusions?

Reviewer #1: Yes

Reviewer #2: (No Response)

3. Has the statistical analysis been performed appropriately and rigorously? 

Reviewer #1: Yes

Reviewer #2: (No Response)

4. Have the authors made all data underlying the findings in their manuscript fully available?

Reviewer #1: No

Reviewer #2: (No Response)

5. Is the manuscript presented in an intelligible fashion and written in standard English?

Reviewer #1: Yes

Reviewer #2: (No Response)

6. Review Comments to the Author

Reviewer #1: My comments were addressed.

My comments were addressed.

My comments were addressed.

My comments were addressed.

Reviewer #2: (No Response)

7. PLOS authors have the option to publish the peer review history of their article (what does this mean?). If published, this will include your full peer review and any attached files.

Reviewer #1: No

Reviewer #2: Yes: Daniel Lapsley

---

## [Editor Report · Acceptance letter]

12 Feb 2020

PONE-D-19-23059R1 

Individual and familial factors predict formation and improvement of adolescents’ academic expectations: a longitudinal study in Sweden 

Dear Dr. Almroth:

I am pleased to inform you that your manuscript has been deemed suitable for publication in PLOS ONE. Congratulations! Your manuscript is now with our production department. 

With kind regards,

on behalf of

Dr. Muhammad A. Z. Mughal 

Academic Editor

PLOS ONE